# Air Pollution, Oxidative Stress, and the Risk of Development of Type 1 Diabetes

**DOI:** 10.3390/antiox11101908

**Published:** 2022-09-27

**Authors:** Katarzyna Zorena, Marta Jaskulak, Małgorzata Michalska, Małgorzata Mrugacz, Franck Vandenbulcke

**Affiliations:** 1Department of Immunobiology and Environment Microbiology, Faculty of Health Sciences, Institute of Maritime and Tropical Medicine, Medical University of Gdańsk, Dębinki 7, 80-210 Gdańsk, Poland; 2Department of Ophthalmology and Eye Rehabilitation, Medical University of Bialystok, Kilinskiego 1, 15-089 Białystok, Poland; 3Laboratoire de Génie Civil et Géo-Environnement, Univ. Lille, IMT Lille Douai, University Artois, YncreaHauts-de-France, ULR4515-LGCgE, F-59000 Lille, France

**Keywords:** particulate matter, selected heavy metals, oxidative stress, type 1 diabetes mellitus, chronic vascular complications

## Abstract

Despite multiple studies focusing on environmental factors conducive to the development of type 1 diabetes mellitus (T1DM), knowledge about the involvement of long-term exposure to air pollution seems insufficient. The main focus of epidemiological studies is placed on the relationship between exposure to various concentrations of particulate matter (PM): PM_1_, PM_2.5_, PM_10_, and sulfur dioxide (SO_2_), nitrogen dioxide (NO_2_), carbon monoxide (O_3_), versus the risk of T1DM development. Although the specific molecular mechanism(s) behind the link between increased air pollution exposure and a higher risk of diabetes and metabolic dysfunction is yet unknown, available data indicate air pollution-induced inflammation and oxidative stress as a significant pathway. The purpose of this paper is to assess recent research examining the association between inhalation exposure to PM and associated metals and the increasing rates of T1DM worldwide. The development of modern and more adequate methods for air quality monitoring is also introduced. A particular emphasis on microsensors, mobile and autonomous measuring platforms, satellites, and innovative approaches of IoT, 5G connections, and Block chain technologies are also presented. Reputable databases, including PubMed, Scopus, and Web of Science, were used to search for relevant literature. Eligibility criteria involved recent publication years, particularly publications within the last five years (except for papers presenting a certain novelty or mechanism for the first time). Population, toxicological and epidemiological studies that focused particularly on fine and ultra-fine PM and associated ambient metals, were preferred, as well as full-text publications.

## 1. Introduction

Air pollution continues to be a global environmental health risk that influences the onset and progression of a variety of health problems, including cardiovascular diseases, lung illnesses, cancer, and central nervous system disorders [1]. According to the 2017 Global Burden of Disease Study, outdoor air pollution caused 4.9 million deaths and 1.4 billion disability-adjusted life years (DALYs) in 2017 [2]. Particulate air pollution, which is primarily composed of organic and elemental carbon, metals, polycyclic aromatic hydrocarbons, inorganic compounds, nitrates, sulfates, and other organic materials (such as polychlorinated biphenyls from industrial manufacturing), is a major component of air pollution and is of great research interest because of its well-documented links to serious short- and long-term adverse health effects [3]. According to the World Health Organization (WHO) estimates from 2018, 91 percent of the global population was exposed to hazardous air (the PM_2.5_ annual average concentration guideline value is 10 g/m^3^). By 2050, it is expected that 68 percent of the world’s population will live in cities, subjecting them to dangerous levels of air pollution on a daily basis [4].

The health impact related to particulate matter (PM) is of major concern to the public and government entities [5]. PM has a high capacity for adsorbing hazardous metals, which can later enter the human body through breathing and cause physiological problems. PM has been associated with a variety of illnesses, including cardiovascular and respiratory illnesses, as well as lung cancer [6]. There is mounting evidence that heavy metals adsorbing on PM are critical to its toxicity and negative health effects. Metal elements adsorbed to PM can also be deposited into soils and water bodies, causing further bioaccumulation in the food chain [7].

Progressive environmental pollution is connected with a dynamic increase of T1DM incidences among children from different countries [8,9,10,11]. Viral infections, low vitamin D levels in infancy, early administration of cow’s milk, excessive weight, and climatic conditions are among environmental factors contributing to T1DM incidences [12,13,14,15,16,17,18]. 

An increasing number of studies confirm a negative relationship between endocrine disruptors and the occurrence of T1DM [18,19]. Particulate Matter (PM) e.g., PM_2.5_ and PM_10_, could be a carrier medium for a wide range of harmful pollutants, including heavy metals or polycyclic aromatic hydrocarbons (PAHs) [20,21]. Fuel combustion processes in the energy and industrial sectors, emissions related to car transport, and home heating are the most prominent sources of pollutant emissions with suspended dust. Atmospheric air is also influenced by pollution sourcing from transport, mainly associated with the process of liquid fuel burning, vehicle operating parts or car tires, and asphalt surface abrasions [22]. Recent studies demonstrated the relationship between air pollution and increased morbidity of respiratory diseases, cardiovascular diseases (CVD), diabetes, increased susceptibility to allergies, and cancer (Figure 1) [8,23]. In this paper, we review available research dealing with a link between air pollution and the development of T1DM.

## 2. Methods

Review of the literature for this review manuscript took place during June 2021–May 2022. Reputable databases, including PubMed, Scopus, and Web of Science, were used to search for relevant literature. Search terms included various combinations of the following key words and phrases: “type 1 diabetes” “onset of type 1 diabetes”, “T1DM”, “T1D”, “particulate matter”, “PM”, “particulate matter metals”, “ambient metals”, “air pollution”, “metal inhalation”, “heavy metal toxicity air pollution”, “PM_2.5_”. Specific heavy metals that have been reported to be associated with particulate matter were also added to searches, including (but, not limited to), “mercury”, “cadmium”, “nickel”, “zinc”, “manganese”, “copper”, and “lead”. Eligibility criteria involved recent publication years, particularly publications within the last five years (except for papers presenting a certain novelty or mechanism for the first time). Population, toxicological and epidemiological studies that focused particularly on fine and ultra-fine PM and associated ambient metals, were preferred, as well as full-text publications. Geographical location of studies was not considered. Exclusion criteria included animal studies that did not use air pollution/inhalation as a source of contamination (e.g., injection with heavy metal), studies that were not fully available online, and studies with a primary focus on type 2 diabetes which only briefly mentioned type 1 diabetes. The rationale for this review paper was to assess recent research examining the association between inhalation exposure to PM and associated metals and the increasing rates of T1DM worldwide.

## 3. Exposure to PM_1_, PM_2.5_, PM_10_, SO_2_, NO_2_, O_3_ and the Morbidity of T1DM

Epidemiological studies indicate the dependence between T1DM development and the concentration of Particulate Matter (PM) with a diameter of 10 microns or less, PM_10_ and PM_2.5_ [23,24,25,26]. In studies conducted in 2015–2016 in the Pomeranian Voivodeship [26], a relationship was found between the average annual concentration of PM_10_ dust (*p* < 0.001), CO (*p* < 0.001), SO_2_ (*p* < 0.001), and a number of new cases of T1DM in children. However, no relationship was found between the mean annual concentration of NO_2_,NO, and the number of new cases of T1DM in children. The results indicated that CO, SO_2_, and PM_10_ dust may be the contributing factors to the development of type 1 diabetes in children and adolescents [26]. A positive correlation between air pollution with PM_10_ and NO_2_ and insulin resistance was also found in studies involving 374 children aged 10 to 18 years [27]. A similar correlation was found in a study involving 397 children aged 10 years exposed to heavy traffic air pollution. The level of insulin resistance was higher in children with greater exposure to air pollution in all models involved—for each 24-s increased exposure to NO_2_ and PM_10_ in the air, an increase of 17.0% and 18.7% in insulin resistance value was observed, respectively [28].

Since heavy metals like cadmium (Cd), (Zn), lead (Pb), copper (Cu), and nickel (Ni), can adsorb to the PM surfaces and contribute to the toxic consequences of PM exposure, the scientists’ attention has also been focused on the importance of trace elements, heavy metals, and metalloids (e.g., Se or Ar) in the development of T1DM [29,30]. Alghobashy A. et al. [29], showed that serum Se, Zn, Mg and Cu were significantly lower in a diabetic group in comparison to the control group. This agreed with Özenç et al. [8], who found lower serum Se and Zn and normal serum Cu levels in children with T1D in comparison to controls [8]. They explained the low serum Se level in patients with T1D might be due to its consumption by the increased activity of the antioxidant GSH-GPx system in order to reduce the free radicals produced by increased oxidative stress [29].

Other, Chiu YM et al. [24], examined associations among prenatal PM_2.5_, maternal antioxidant intake, and childhood wheeze in an urban pregnancy cohort (*n* = 530). Daily PM_2.5_ exposure over gestation was estimated using a satellite-based spatiotemporally resolved model. Mothers completed the modified food frequency questionnaire. Average energy-adjusted percentile intake of β-carotene, vitamins (A, C, E), and trace minerals (zinc, magnesium, selenium) constituted an antioxidant index (AI). Higher AI was associated with decreased wheeze in black babies (OR = 0.37 (0.19–0.73), per IQR increase). BDLIMs identified a sensitive window for PM_2.5_ effects on wheeze among boys born to black mothers with low AI (at 33–40 weeks gestation; OR = 1.74 (1.19–2.54), per µg/m^3^ increase in PM_2.5_). Relationships among prenatal PM_2.5_, antioxidant intake, and child wheeze were modified by race/ethnicity and sex [24].

In 2019, a study evaluating the impact of the traffic-related service on blood pro-inflammatory biomarkers in adolescents with type 1 diabetes was published. The conducted study showed an increase in the level of IL-6 by 8.3% (95% CI: 2.7%, 14.3%) and in the level of hs-CRP by 9.8% (95% CI: 2.4%, 17.7%) after short-term exposure to elemental carbon (EC). Air pollution has also been shown to increase the risk of cardiovascular disease in adolescents already suffering from T1DM [31]. A study of 13,948 children aged 0–9 found 15 out of 53 demographic and environmental risk factors, including PM_10_, NO_2_, NO, CO, and Pb in soil, the presence of radon (Rn), outdoor lighting at night, overpopulation, population density, and ethnicity as significantly associated with type 1 diabetes [32].

Other studies showed the correlation between the occurrence of T1DM and the density of the paved roads network, the annual emission of atmospheric pollutants from stationary sources, and the number of buses per 100,000 inhabitants [33]. One of the teams [34] assessed the exposure of children and adolescents suffering from type 1 diabetes to the mean annual concentration of PM_10_, NO_2_ particulate matter, and the long-term concentration of O_3_ in the atmospheric air (AOT40 index). The research was carried out on a group of 771 patients aged 11–21 years. The tests did not reveal the effect of suspended dust PM_10_ and nitrogen dioxide NO_2_ on the level of HbA1c (mmol/mol). It was found that the long-term ozone concentration in the air (AOT40) was inversely correlated with HbA1c (mmol/mol) in the blood plasma (IQR estimate: −1.86; 95% CI: (−3.27; −0.44); *p* = 0.01). No correlation was found between the average annual concentration of PM_10_ suspended dust, the concentration of NO_2_ and the average five-year concentration of ozone (AOT40 index), and the daily dose of insulin [U/kg body weight]) [34]. In long-term studies conducted on a group of 37 372 people with type 1 diabetes aged < 21 years, no correlation was found between the mean annual exposure to PM_10_ particulate matter and the level of glycosylated hemoglobin (HbA1c) [35]. There was also no correlation between the average annual NO_2_ concentration in the atmospheric air and HbA1c. However, researchers found an inverse correlation between the five-year average ozone concentration in the air (AOT40 index) and HbA1c [35]. The same team [36] also conducted studies on a group of 31,131 patients with type 1 diabetes, aged < 21 years. Researchers used a mixed model to investigate the relationship between HbA1c levels and the average annual concentration of PM_10_, NO_2_, and the cumulative exposure to O_3_-AOT ozone in the ambient air. The model was adapted to gender, age, diabetes duration, year of treatment, migration, type of insulin treatment, degree of urbanization, and socioeconomic index. Higher levels of HbA1c with higher mean annual concentrations of PM_10_ and NO_2_ in the atmospheric air were observed. An inverse relationship between the five-year mean concentration of O_3_-AOT and HbA1c was also found. Differences in HbA1c levels between air pollution quartiles were small but statistically significant.

It should be mentioned that pregnant women exposed to polluted air can give birth to children susceptible to developing type 1 diabetes [37,38]. A positive relationship indicated between exposure to mean annual O_3_ concentration in the first trimester of pregnancy and diabetes occurrence in children (hazard ratio (HR) for interquartile rise (IQR) = 2.00, 95% CI: 1.04–3.86) has been observed [39]. Studies conducted on Wistar rats have proven the harmful effect of PM_10_ suspended dust in the air during the prenatal period [40]. In that study, pregnant rats received a dose of 50 µg PM/day until the end of the lactation period. After birth, the biochemical parameters of the mothers, as well as their male offspring at 21 and 90 days of age, were analyzed. It has been shown that exposure to PM during perinatal life in rats leads to glucose dyshomeostasis in male offspring, both early and later in life [40]. Another interesting experiment evaluated the effect of Diesel Exhaust Particles (DEP), as the main component of PM_2.5_ dust and ultrafine (nano) particles smaller than (≤0.1 µm), in mice with type 1 diabetes induced by streptozotocin [41]. In diabetic-induced mice, after administration of DEP (0.4 mg/kg), an increase was recorded in the activity of pancreatic amylase and markers of oxidative stress, including 8-isoprostane, superoxide dismutase. At the same time, the level of glutathione decreased. Table 1 summarizes the effect of chemical air pollution on the risk of developing type 1 diabetes.

This section may be divided by subheadings. It should provide a concise and precise description of the experimental results, their interpretations, as well as the experimental conclusions that can be drawn.

## 4. The Risk of T1DM Development Associated with Exposure to Selected Heavy Metals

Particulate matter is made up of inorganic and organic elements, as well as nitrates and sulfates. By adsorbing to the surface of ambient PM, particularly PM_2.5_, viruses, bacteria, volatile organic compounds (VOCs), and heavy metals can be carried [9,10,49,50]. Due to growing car use and traffic congestion, traffic emissions are a major source of heavy metal pollution in the environment [9]. Automobile emissions, for example, discharge a variety of heavy metals into the environment, including Cu, Zn, Cd, As, mercury (Hg), Mn, cobalt (Co), and iron (Fe). Cu and Zn can also be released into the air as a result of tire abrasion, lubricants, and vehicle corrosion, whilst Cd contamination is caused by aged automotive tires, gasoline consumption, and car body and brake lining wear. Along with power plants, mining, metal smelting, and chemical plants, industrial activity contributes to particle-bound heavy metal air pollution [10]. Construction activities, such as building demolition and reconstruction, pesticide and fungicide spraying, and residential and commercial heating are all sources of PM-associated heavy metal pollution [21]. Several studies have found high levels of adsorbed hazardous heavy metals from both natural and anthropogenic sources, which can be harmful to human health. Metals adsorbed to PM vary in type and concentration depending on the source and location. Manganese (Mn), zinc (Zn), iron (Fe), cadmium (Cd), copper (Cu), arsenic (As), barium (Ba), lead (Pb), aluminum (Al), and nickel (Ni) are often significant metal components of PM_2.5_ [10,50]. On the other hand several metals are essential components of biological functions, a reduced amount of which in the body may result in deficiency syndromes, while their higher concentrations may be toxic. Toxicity is determined by several factors, including dose, route of exposure, age, sex, and even nutritional status [51,52]. Non-essential metals, including As, Cd, Cr, Pb, and Hg, are priority metals of importance for public health, due to their high degree of toxicity, even at low doses [52]. They cause multi-organ damage and are considered systemic toxic substances, even at lower exposure levels. In addition, these metals are classified as known or probable human carcinogens, according to the U.S. Environmental Protection Agency (EPA) and the International Agency for Research on Cancer (IARC) [52]. Cd, As, Co, Hg, Mn, and Pb are documented to be endocrine disruptors [46,53].

**Cadmium (Cd)** is one of the most common heavy metals in nature. It occurs in air, water, and soil. The main sources of environmental pollution with cadmium are lead–zinc ore mines, zinc and lead smelters, and electroplating plants [54]. Cadmium accumulates in the human body in the liver, bones, kidneys, pancreas, thyroid, and hair [55]. Smoking can also be an important source of cadmium in the human body [56]. In a study on the cord blood of children, it was shown that the umbilical cord blood of children who in later years developed type 1 diabetes mellitus had an increased concentration of (Al), (Hg), and (As) (*p* = 0.006), compared to the control. Thus, exposure to toxic metals during pregnancy may be one of several environmental factors contributing to the subsequent disease process [57]. 

**Vanadium (V)** is the 22nd most abundant element on earth (0.013% *w*/*w*), and it is widely distributed in all organisms. In humans, the vanadium content in blood plasma is around 200 nM, while in tissues it is around 0.3 mg/kg and is mainly found in bones, liver, and kidney. In vertebrates, vanadium enters the organism principally via the digestive and respiratory tracts through food ingestion and water, and air inhalation [58]. So far, the authors have shown that vanadium administered in the drinking water to streptozotocin (STZ) diabetic rats restored elevated blood glucose to normal. Subsequent studies have shown that vanadyl sulfate can lower elevated blood glucose, cholesterol and triglycerides in a variety of diabetic models. In the BB diabetic rat, a model of insulin-dependent diabetes, vanadyl sulfate lowered the insulin requirement by up to 75% [58].

Galvez-Fernandez et al. [59] analyzed the association of 11 metals (urine antimony, arsenic, barium, cadmium, chromium, cobalt, molybdenum, vanadium, and plasma copper, selenium and zinc) with metabolic patterns, and the interacting role of candidate genetic variants, in 1145 participants, a population-based sample from Spain. Exposures to cobalt, plasma copper, selenium, zinc, and arsenic, but not to vanadium, were associated with several metabolic patterns involved in chronic disease [59]. Vanadium compounds have been primarily investigated as potential therapeutic agents for the treatment of various major health issues, including cancer, atherosclerosis, and diabetes. The translation of vanadium-based compounds into clinical trials and ultimately into disease treatments remains hampered by the absence of a basic pharmacological and metabolic comprehension of such compounds [59,60].

**Zinc****(Zn)** is a micronutrient of which the daily requirement varies, depending on age, in the range of 3–5 mg/day for infants, 10 mg/day for children, and 10–15 mg/day for adults. This micronutrient is taken in by the alimentary route, and its total content in the human body is within 1.5–4 g, depending on the source. High zinc content has been determined in many tissues; however, these concentrations vary considerably [61]. Zn is involved in the metabolism of proteins and carbohydrates. It is also required for the proper functioning of the circulatory, skeletal and reproductive systems. Ions of this metal are involved in the synthesis and regulation of insulin secretion and signaling, as well as glucagon secretion and the secretion and activation of pancreatic digestive exoenzymes. Zn also plays an important role in the functioning of pancreatic β-cell islets (the concentration has been estimated at up to 20 mg), where it is stored in secretory vesicles in a complex with insulin. Studies carried out on rats have proven the relationship between zinc deficiency and a decrease in insulin secretion and activity, and, thus, lower glucose uptake [61]. The results of model studies of oral zinc supplementation in type 1 and 2 diabetes mellitus in rats showed that it reduces insulin resistance, elevated insulin levels, and the cytotoxic effect of streptozotocin. The protective effect of zinc was found to probably be due to the reduction of oxidative stress in the kidney tissue of diabetic rats in [62].

Forte et al. [25] investigated the relationship between the occurrence of type 1 and type 2 diabetes and the concentration of heavy metals in the blood. The researchers studied 192 patients diagnosed with type 1 diabetes, 68 with type 2 diabetes, and 59 as a control subjects. Type 1 diabetes was found to be associated with Cr (*p* = 0.02), Mn (*p* < 0.001), Ni (*p* < 0.001), Pb (*p* = 0.02), and Zn (*p* < 0.001) deficiency, and type 2 diabetes with Cr (*p* = 0.014), Mn (*p* < 0.001), and Ni (*p* < 0.001) deficiency. Furthermore, in type 1 diabetes, there was a positive correlation between Pb and age (*p* < 0.001, ρ = 0.400) and Pb and BMI (*p* < 0.001, ρ = 0.309), while a negative correlation between Fe and age (*p* = 0.002, ρ = −0.218). In type 2 diabetes, there was a negative correlation between Fe and age (*p* = 0.017, ρ = −0.294) and Fe and BMI (*p* = 0.026, ρ = −0.301) [25]. 

Similar results were obtained by a team examining the content of micronutrients in the blood of 192 people with diagnosed type 1 diabetes [63]. The concentration of Zn in the blood of women was 5975 µgL^−1^ (*p* < 0.0001), and it was significantly lower than in the studied population of men [63]. The studies also showed that there was no correlation between the amount of zinc in the blood and glycosylated hemoglobin (HbA1c). A decrease in the concentration of Se, Zn, Mg, and Cu in blood plasma, as well as reduced GSH glutathione and GPx glutathione peroxidase, was detected in erythrocytes of children with uncontrolled type 1 diabetes [27]. Researchers showed a negative correlation between the concentration of Se, Zn, Mg, GSH, and GPx levels in plasma and erythrocytes, and glycated hemoglobin HbA1c. Moreover, they showed a positive correlation between the plasma concentration and the GSH and GPx concentration in erythrocytes ([r = 0.56, *p* < 0.001], [r = 0.78, *p* < 0.001], respectively) [27]. In studies involving 26 patients with T1DM and 80 patients with T2DM, it was confirmed that the plasma levels of Zn and the Zn/Cu ratio were approximately 3- and 2-fold lower than in the control group studied [64]. There were no significant differences in plasma Cu levels. It was further found that decreased levels of Zn and increased levels of oxysterol were significantly associated with HbA1c and fasting plasma glucose levels. In studies on the content of micronutrients in the blood of 88 patients (mean age 16.5 years) with type 1 diabetes, there were no differences in the concentrations of Zn and Cr (*p* = 0.153 and 0.515, respectively) in the blood in correlation to the control group. There was also no correlation between chromium levels and HbA1C, duration of diabetes, and insulin dose/BMI (*p* > 0.05). However, a significant difference was found between zinc levels and insulin dose/BMI (*p* = 0.043) [65,66]. Studies of the concentration of Mg, Zn, Cu, Se, and HbA1c in 45 patients with type 1 diabetes and 54 patients with type 2 diabetes were conducted in [66]. The researchers did not observe a significant difference in the total plasma zinc concentration between patients with T1DM and T2DM and their controls. The plasma zinc concentration of patients did not correlate with the concentration of HbA1c in patients with diagnosed diabetes T1DM, nor in those with diagnosed diabetes [66]. However, a statistically significant increase of this element in the blood of type 1 diabetes was noticed in [67]. The metabolic relationship between copper and zinc has also been widely documented. It has been confirmed that long-term intake of doses of zinc much higher than the daily requirement may lead to disorders of copper absorption, which, in turn, lead to impaired iron utilization and heme synthesis, and, thus, to anemia [61]. Research on the chemical content of drinking water shows that low levels of zinc increase the risk of developing diabetes T1DM [68]. Kidney tissue antioxidant enzyme activities, which were significantly impaired in the untreated diabetic group, were reversed in zinc treated diabetic groups, thus, showing the beneficial effect of Zn treatment in diabetes via its antioxidative effects [69]. In the study by Singh et al., a significantly lower glucose elevation in Cd-exposed animals was detected [70]. Many studies have shown that the metal concentration in the blood significantly affects the duration of type 1 diabetes and the degree of metabolic control [71,72,73,74,75,76,77,78].

**Copper (Cu)** content in the human body is low and amounts to approximately 150 mg. Most of this element accumulates in the liver, muscles, bones, and brain. In blood plasma, 95% of copper is combined with a specific protein, ceruloplasmin, and only a small amount is associated with albumin and the β-globulin fraction. In red blood cells, 80% of the copper is contained in superoxide dismutase (SOD), and the rest of it is stored as complexes with a protein called erythrocuprein [75]. The biological role of copper is related to its participation in the structures and functions of many enzymes (tyrosinase, oxidase, cytochrome C, superoxide dismutase). In addition, it is actively involved in the synthesis of collagen and elastin, and myelin formation, and it affects osteogenesis and erythropoiesis. On the other hand, disorders of copper metabolism, as a result of a genetic defect, may cause its pathological accumulation in organs and tissues (Wilson and Menkes disease) [28,76,77]. Studies on the content of Cu in the blood plasma in [67] did not show a higher level of this element in patients with type 1 diabetes compared to a healthy control group, and showed no relationship between the concentration of Cu, gender, glycemic control or the presence of microalbuminuria and trace element plasma levels in patients with type 1 diabetes. However, the research team of Peruzzu et al. received different results in [63]. Scientists have shown that the concentration of copper in the blood of women diagnosed with type 1 diabetes was statistically higher (*p* < 0.0001) than in the studied men. Moreover, the copper content in the blood of the studied patients was positively correlated with HbA1c. There were also observed statistically significant higher correlations between the concentration of copper in the group of men with well and poorly controlled diabetes [63]. In the blood plasma of patients with T1DM diabetes, a negative correlation was found between Cu: Zn and the concentration of HbA1c. Moreover, a positive correlation was found between the HbA1c concentration and the copper concentration in the group of men studied in the blood plasma in [66]. Another team found high total plasma levels of copper and ceruloplasmin in T1DM patients compared to the control group [28]. The conducted studies included 63 adult patients with T1DM and 65 healthy individuals [28]. Studies on the concentration of Mg, Mn, Zn, Se, and Cu in the plasma, and the concentration of zinc-copper superoxide dismutase (CuZn-SOD), catalase (CAT), and glutathione peroxidase (SeGSH-Px) in erythrocyte hemolysates in T1DM patients and their siblings showed that, compared to the control group, children with T1DM had lower plasma levels of Mg and Zn, lower zinc-copper superoxide dismutase (CuZnSOD) and higher levels of Cu. Their siblings had a lower concentration of Zn in the blood plasma. Children with type 1 diabetes and their siblings also had higher catalase activity (CAT) and lower total antioxidant status (TAS). No differences were found between the concentration of Mn and Se in the plasma in the studied children. The conducted studies also showed that the intensification of oxidative stress in T1DM was accompanied by changes in the activity of antioxidant enzymes and non-enzymatic antioxidant defense mechanisms (TAS) [71].

**Lead (Pb)** is another toxic element that may influence the development of T1DM diabetes is lead. The absorption of lead occurs mainly through the respiratory system and the gastrointestinal tract. Approximately 30–40% of inhaled lead is absorbed into the blood. In contrast, absorption through the digestive system varies, depending on nutritional status and age. Lead is especially dangerous for children. Following contact with lead, they develop abdominal pain, vomiting, weight loss, anemia, and symptoms of renal failure. Even small doses of this element cause distraction, hyperactivity, and excessive irritability. High doses of lead permanently damage the brain and can even be fatal [3,78]. In studies conducted in patients with type 1 and type 2 diabetes, a significantly lower concentration of lead in the blood was found compared to the control group [25]. Differences in blood lead content were found in the studied patients in the age range from 18 to 40 years (12.4 ng/mL) and in the age group 41–60 years (21.9 ng/mL). The demonstrated dependence may be reflected in the mechanism related to the aging of the organism when an increasing amount of lead, which has accumulated in the bones, is released into the circulatory system. Comparing the duration of developing type 1 diabetes with blood lead levels, higher blood lead levels were found in patients who had type 1 diabetes for less than 10 years [25].

**Chrome (Cr)** daily dose for humans has been set at about 50 μg. The daily consumption of this element may vary widely and sometimes reaches 4 times higher levels than the proposed necessary dose. Studies conducted on patients diagnosed with type 1 diabetes have shown that Cr is positively correlated with HDL cholesterol (*p* = 0.0079) in men and with fasting plasma glucose (FPG) (*p* = 0.0149) [63]. In studies conducted on 12 patients with impaired fasting glucose (IFG), 15 patients with IGT, 25 patients diagnosed with T1DM diabetes, 137 patients with T2DM, and 50 healthy individuals, the levels of Cr and Fe in plasma and urine of patients with IFG, impaired glucose tolerance (IGT), type 1 diabetes and type 2 diabetes were compared [73]. No significant differences in plasma Fe levels were found in all groups studied. The results of the research also showed that the levels of Cr and Fe in the urine were highest in patients with T1DM diabetes. The obtained results may suggest a potential role of these elements in the pathogenesis of diabetes [73]. The results of model studies on rats indicated a beneficial effect of organic and inorganic chromium compounds on carbohydrate and lipid metabolism. It was found that dietary supplementation with chromium decreased insulin concentration in the blood of healthy rats and improved insulin sensitivity, and decreased glycosylated hemoglobin (HbA1c) and glucose levels in animals with induced diabetes, insulin resistance, or metabolic syndrome. In animal studies, it has also been shown that chromium may be hypolipemic. Under the influence of dietary supplementation with various compounds of this element, a decrease in total cholesterol, LDL fraction, and triglycerides, and an increase in β-oxidation of fatty acids, in both healthy and sick rats were found [79]. In studies of the therapeutic and antioxidant effects of chromium picolinate (CrPic) on rats (DN rats) with diabetic nephropathy (intraperitoneal injection of streptozotocin), it was found that eight weeks of CrPic supplementation at 1 mg kg^−1^ d^−1^ improved renal function and reversed pathological changes, such as interstitial fibrosis or glomerular sclerosis. Inhibition of the expression of TGF-β1 and the proteins Smad2 and Smad3 was also observed [74].

**Nickel (Ni)** is a ubiquitous trace metal found in soil, and water, including drinking water, atmospheric air, and the biosphere [2,80,81]. There is about 0.5 mg Ni/70 kg in the adult human body, which is 7.4 μg/kg. The physiological role of nickel is to activate certain enzymes (including tyrosinase, arginase, deoxyribonuclease), increase hormonal activity, stabilize nucleic acid structures, and participate in lipid metabolism. Nickel is an indispensable component of bacterial enzymes. Furthermore, nickel ions act antagonistically to calcium ions which, by activating calmodulin, start the metabolic pathway of arachidonic acid. Depending on the time and dose of exposure to Ni by inhalation, asthma, cardiovascular diseases, increased risk of lung cancer, and upper respiratory tract cancer may be diagnosed. Nickel is also a very common factor of contact dermatitis [2,81]. Studies of the concentration of Ni in the blood plasma of patients diagnosed with type 1 and type 2 diabetes showed deficiencies of this metal in comparison with the control group [25]. In the elderly population, aged over 60 years, it was shown that the concentration of Ni in the blood plasma was lower and showed a statistically significant difference compared to non-diabetic individuals [25]. In studies on the content of micronutrients in the blood of children with type 1 diabetes, an inverse correlation was found between the level of HbA1 and the concentration of nickel in the blood plasma in children with poor metabolic control (r = 0.6) and a weak inverse correlation in the group of moderately balanced patients (r = 0.34) [72]. Children were divided according to the level of HbA1c, following the recommendations of the *International Society for Pediatric and Adolescent Diabetes* (ISPAD), into three groups: HbA1c < 7.5% in the well-controlled group, HbA1c > 7.5–9 in the moderately controlled group, and HbA1c > 9 patients misaligned. In the group of children with proper metabolic control of diabetes, significantly higher levels of nickel in the blood plasma were found than in children with poor metabolic control. Moreover, the statistical analysis of the studied parameters showed that the nickel concentration in erythrocytes in less well-balanced children was much lower than in children without diabetes. The study showed that the duration of type 1 diabetes and the degree of metabolic control of diabetes significantly affected the values of nickel concentration in the blood [72]. Table 2 shows exposure to selected heavy metals and the possible risk of developing type 1 diabetes.

## 5. Effects of Heavy Metals, Particulate Matter on the Oxidative Stress, and Gut Microbiome

Air pollution causes oxidative stress in the body, which is a known risk factor for metabolic dysfunction illnesses [30]. Oxidative stress can lead to lipid peroxidation, antioxidant depletion, and pro-inflammatory signaling activation, all of which can trigger a chain of biological events that affect distant organs. Cancer, atherosclerosis, hypertension, ischemia/perfusion, diabetes, and asthma are only a few of the diseases it causes [82]. The links between air pollution exposure and an elevated risk of diabetes have a strong biological basis. Inhaling air pollutants, such as PM_2.5_, can cause systemic effects, including autoimmune responses and metabolic dysfunctions [39]. Although the specific molecular mechanism(s) behind the link between increased air pollution exposure and a higher risk of autoimmune and metabolic dysfunction-related disorders is yet unknown, available data indicates air pollution-induced inflammation and oxidative stress as a significant pathway [83].

Exposure to fine particulate matter (PM_2.5_) air pollution originating from the combustion of fossil fuels is closely linked to the induction of both systemic inflammation and oxidative stress among the numerous air pollutants [84]. The postulated mechanism is that fossil fuel particulate matter, particularly coal combustion PM_2.5_, is high in both metals and sulfur which can produce oxidative stress, and sulfur, which causes acidity, increasing the bioavailability of reactive metals and causing systemic damage [84]. By creating reactive oxygen species (ROS), such as superoxide radicals, hydrogen peroxide, and nitric oxide, toxic metals, As, Cd, Hg, and Pb, can cause oxidative stress. Many metals have been found to enhance lipid peroxidation, or the free radical-driven oxidative alteration of low-density lipoprotein (ox-LDL), a well-known early cause of atherosclerosis [85]. By oxidative pathways, Cd can damage vascular tissues, cause endothelial dysfunction, and promote atherosclerosis. Pb is known to cause the generation of reactive oxygen species (ROS), and Pb-induced oxidative stress can result in protein, nucleic acid, and lipid peroxidation [86]. Cu and Zn imbalances, for example, are required for balanced oxidant-antioxidant pathways, and Cu and Zn imbalances can increase vulnerability to toxic metal-induced oxidative damage to islet cells, leading to insulin resistance pathogenesis. Selenium (Se) is a cofactor of the antioxidant enzyme glutathione peroxidase, which helps to reduce oxidative stress caused by Cd/Pb [87]. The impact of selected heavy metals commonly seen in PM on β-cells and the development of T1DM is presented in Figure 2.

In addition to oxidative stress, the influence of heavy metals on the gut microbiome also plays a role. Previous studies have shown that exposure to heavy metals adsorbed on PM has a significant effect on the intestinal microbiome, leading to dysbiosis of the physiological bacterial flora [51,88,89]. To determine how particulate matter affect the intestines and gut microbiome, wild-type (WT) mice were exposed orally to PM_10_ for 7 or 14 days. In turn, to assess longer term effects of exposure, IL10 deficient (−/−) mice were subjected to the same treatment for 35 days [88]. WT mice exposed to PM_10_ for a short duration had alteration in immune gene expression, increased gut leakiness, and enhanced pro-inflammatory cytokine secretion into the small intestine. In other words, a significant increase in IL-17, IL-1β, TNFα, IL-12 and IL-13 was seen in the colon of IL-10−/− mice treated with PM_10_. PM_10_ induced increased IL-17 and IL-13 in the colons of WT mice, and these animals demonstrated significant changes in the relative amounts of *Bacteroidetes* spp., *Firmicutes* spp., and *Verrucomicrobia* spp. [88].

In other studies, the effect of lead on bacterial microflora in experimental mice was presented. Non-agouti (a/a) offspring derived from A *^vy^* /a male mice bred to a/a female mice exposed, from gestation through lactation, to Pb (32 ppm in the drinking water) underwent shifts in gut microbiota populations with *Bacteroidetes and Firmicutes* inversely associated with maternal Pb exposure. Cultivable aerobes decreased but anaerobes increased in the Pb-exposed offspring. Intestinal flora changes were associated with an increase in adult body weight in males but not in females [51].

Arsenic is ubiquitous in nature, highly toxic, and, in some countries, contamination in drinking water has been shown to be above the WHO and EPA recommended limit of 10 μg/L [89]. In a recent study, the authors examined stool samples (*n* = 42) collected from members of the Mahuawa (*n* = 20) and Ghanashyampur (*n* = 22) communities in southern Nepal [89]. The 16S rRNA gene was amplified from fecal samples using Illumina-tag PCR and subjected to high-throughput sequencing to generate the bacterial community structure of each sample. Bioinformatics analysis and multivariate statistics were conducted to identify if specific fecal bacterial assemblages and predicted functions were correlated with urine arsenic concentration. The conducted results revealed unique assemblages of arsenic volatilizing and pathogenic bacteria positively correlated with increased arsenic concentration in individuals within the two respective communities. Additionally, they observed that commensal gut bacteria negatively correlated with increased arsenic concentration in the two respective communities. The study’s authors revealed that arsenic has the potential to debilitate healthy humans by contributing to disorders like heart and liver cancers and diabetes, and it has already been shown to contribute to serious diseases and disorders, including skin lesions, gangrene and several types of skin, renal, lung, and liver cancers in disadvantaged areas of the world, like Nepal [89].

Interesting research was presented by the team of Griggs JL et al. [90]. In recent studies, the authors explored the relation between arsenic exposure and changes in the composition of the gut microbiome and in arsenic bio-accessibility. Here, they used a simulated GI model system (GIMS) containing a stomach, small intestine, colon phases and microorganisms obtained from mouse feces (GIMS-F) and cecal contents (GIMS-C) to assess whether exposure to arsenic-contaminated soils affected the gut microbiome and whether composition of the gut microbiome affected arsenic bio-accessibility. Soils contaminated with arsenic did not alter gut microbiome composition in the GIMS-F colon phase. In contrast, arsenic exposure resulted in the decline of bacteria in GIMS-C, including members of *Clostridiaceae*, *Rikenellaceae*, and *Parabacteroides*. Arsenic bio-accessibility was greatest in the acidic stomach phase of GIMS (pH 1.5–1.7); except for the GIMS-C colon phase [90]. In other study, in 6- to 8-week-old C57Bl/6 Tac male mice exposed for 2, 5, or 10 weeks to 0, 10, or 250 ppb arsenic, time and dose-dependent effects on the gut microbial community were found, especially for *Bacteroidetes* and *Firmicutes* [91]. Arsenic treatment removed the bacterial biofilm residing along the mucosal lining and altered the diversity and abundance of microorganisms, with bacterial spores increasing and intracellular inclusions reducing with the 250 ppb dose [91].

In the present manuscript, only a few examples are given of the effects of heavy metals from soil, water, and air on the gastrointestinal microbiome, as this issue is beyond the subject covered in the manuscript.

## 6. Air Pollution and Chronic Vascular Complications

Untreated or poorly controlled diabetes can increase the risk of chronic complications, the treatment of which places a huge burden on healthcare systems around the world [92]. Multidisciplinary studies available for many years have shown that chronic diabetic complications develop as a result of progressive vascular damage of both small and large blood vessels. One of the major chronic complications of diabetic patients is diabetic retinopathy (DR), the leading cause of blindness worldwide. Despite extensive research, the complex pathogenesis of diabetic retinopathy has not been fully elucidated [93,94,95]. For many years it was believed that DR only manifested itself in microangiopathic changes, which were entirely responsible for the loss of vision in diabetic patients. In the light of current knowledge on microangiopathic changes in the fundus, diabetic retinopathy is perceived as a neurodegenerative disease [96]. There are studies showing that in patients with DM higher particulate matter exposure can lead to higher risk of DR [96]. Moreover, a retrospective cohort study showed that atmospheric air pollution, including particulate matter, PM_2.5_ PM_10_, NO_2_, SO_2_ and ozone O_3_, was associated with an increased risk of central retinal artery obstruction (CRAO), especially in patients with diabetes or hypertension and people over the age of 65 [97]. Apart from changes in the fundus, nephropathy was also observed in the course of diabetes [98]. In light of recent studies in diabetic patients, deterioration of renal function and end-stage renal disease may also be associated with exposure to PM in ambient air. For example, the long-term effect of CO and PM_2.5_ exposure on microalbuminuria in type 2 diabetes has been investigated. Over 800 patients with type 2 diabetes were enrolled in the study. Urinary albumin to creatinine ratio (ACR) was recorded annually for 10 years. Exposure to air pollutants CO, NO_2_, O_3_, SO_2_ and PM_2.5_ was interpolated from 72 air quality monitoring stations. The authors showed that exposure to high levels of CO and PM_2.5_ increased albuminuria in type 2 diabetes [99].

Exposure to particulate matter (PM) and gaseous air pollutants has also been linked to macrovascular diseases, including cerebrovascular and cardiovascular diseases, progression of coronary calcification and acceleration of atherosclerosis [80,81]. The beforementioned study also showed that exposure to high concentrations of airborne dust, even for short periods, is associated with an increased risk of acute cardiovascular events (myocardial infarctions and strokes). Constant exposure also increased the risk of chronic cardiovascular disease [100]. Cardiovascular disease (CVD) is the leading cause of death among people with type 1 diabetes (T1D), who have a 10–29 times greater risk of dying from CVD compared to healthy controls. Atherosclerosis can start as early as childhood and develops more aggressively in adolescents with T1D [101]. The etiopathogenesis of cardiovascular disease is complex, with genetic factors, lifestyle factors, including eating habits, and environmental factors, including air pollution, undoubtedly contributing to its development [102,103,104]. A meta-analysis of 11 studies (data of 33,922 people in total) confirmed that the concentration of PM_2.5_ correlates more strongly than the concentration of PM_10_ with carotid intima-media thickness (CIMT). An increase in the concentration of suspended dust by 10 µg/m^3^ is associated with a significantly greater thickness of CIMT, for the PM_2.5_ fraction, by 16.79 µm, and, for the PM_10_ fraction, by 4.13 µm. The magnitude of the impact of exposure to particulate matter and indicators of subclinical atherosclerosis depend on gender and the relationship is clearly greater in women [102].

The relationship between gas pollution and worsening of heart failure was even clearer. An increase in carbon monoxide concentration by 1 ppm (parts per million) increased the risk of hospitalization by 3.52%, and sulfur dioxide by 2.36%, and nitrogen dioxide by 1.7%. Only the relationship between ozone concentration and exacerbation of heart failure was not found. The authors estimated that the reduction of PM_2.5_ dust pollution by an average of 3.9 µg/m^3^ in the United States could prevent 7978 hospitalizations due to worsening heart failure annually [105].

Other studies have examined associations between traffic-related air pollutants (NO_2_), carotid atherosclerotic plaque, and cardiometabolic disorders associated with cardiovascular disease. Cross-sectional analyses were conducted among 2227 patients from the Stroke Prevention and Atherosclerosis Research Centre (SPARC) in Canada, Ontario, London. The authors showed that NO_2_ was positively associated with triglycerides, total cholesterol, and the ratio of low- to high-density lipoprotein cholesterol. The authors demonstrated that even low levels of traffic-related air pollutants were linked to atherosclerotic plaque burden, an association that might be partially attributable to pollution-induced diabetes mellitus [106].

In other studies on the impact of air pollution on heart failure, it was found that an increase in the concentration of PM_2.5_ suspended dust increased the risk of hospitalization for heart failure by 2.12% for each increase in the concentration of this dust by 10 µg/m^3^ [107].

The long-term exposure to PM_2.5_ associated with cardiovascular risk factors in a less educated Chinese population was assessed in another study. In the study, a total of 19,236 participants from the Chinese Physiological Constant and Health Condition survey were included, of which nearly half were male (47.0%). The association between long-term PM_2.5_ exposure and cardiovascular risk factors was demonstrated. PM_2.5_ was associated with a higher prevalence of diabetes, hypertension, and being overweight in a less-educated population [108]. Table 3 presents selected studies on air pollution, including heavy metals and complications.

## 7. Modern Methods for Air Pollution Monitoring

Due to the harmful effects of air pollution on human health and the growing population of large cities, extensive monitoring of air pollution levels is necessary [109,110,111,112]. A Report from 2019 entitled, “*State of Global Air/2019*—*A special report on global exposure to air pollution and its disease burden*” [109], by the *Health Effects Institute* and *Institute for Health Metrics and Evaluation’s Global Burden of Disease Project*, drew the world’s attention towards the need for adequate quality monitoring. According to the aforementioned report, air pollution constituted the fifth most important risk factor for death from all causes, for all ages and both genders, with approximately 4.9 million deaths. The World Health Organization (WHO) estimates that 90% of people breathe polluted air globally [110]. Breathing air containing suspended particles of diameters less than 2.5 μm increases the risk of developing heart disease, chronic respiratory diseases, lung infections, and cancer, as well as the entire spectrum of diseases of immune and psychological backgrounds [109]. As predicted by the United Nations (UN), the population of cities will increase up to 2.5 billion by 2050, thus steps leading to reducing air pollution must be taken immediately. With the introduction of air quality guidelines, the WHO implemented versified tools, e.g., AirQ, Health Economic Assessment Tool, (HEAT), or Integrated Transport and Health Impact Modeling Tool (ITHIM) [4,109].

As a result of growing awareness of the health consequences of exposure to polluted air, government regulations more vigorously support the effective monitoring and control of air pollutants. One of the international initiatives, called C40 *Cities Climate Leadership* (C40) [110], brings together 97 cities from around the world focusing on taking measures to reduce greenhouse gas emissions and counteracting negative climate changes, thus improving the quality of their inhabitants’ life. As a part of the C40 group, in 2018, the mayor of London, Sadiq Khan, initiated the implementation of a groundbreaking project of air pollution monitoring networks. The London network, based on city-wide sensors, was intended to be one of the most modern in the world [111].

There is a growing demand for air quality monitoring systems in global markets and it is estimated that by 2025 this market will reach $6 billion [112]. Traditional measurement stations are expensive and stationary and, thus, the possibility of monitoring wide fragments of areas is limited. Therefore, traditional measuring stations are more often replaced by cheap and simple micro-sensors, which are used to further map the air quality in high-resolution, using geostatistical methods [113]. The integrated mobile platform for air pollution measurement and registration may be installed on a drone to assure maximum mobility [114], which allows for measurement directly from a source of contamination with simultaneous image registration. Moreover, the measurement of specific types of emissions is possible; for instance, from combustion processes of hazardous substances. Using the LTE or WiFi network, the collected data can be easily transferred to interested recipients and mobile devices.

The possibility of utilization of multiple sensors for air quality monitoring in real-time at high temporal and spatial resolutions is currently a developing direction in mass air quality monitoring [115]. However, these measurements have several uncertainties, including instability and measurement disturbances due to weather conditions and/or interfering compounds [116].

A pilot project involves the utilization of a sensor for “Intelligent and Scalable Air Quality Monitoring” combined with a 5G connection [117]. The platform was proposed by researchers from Korea [118]. Thirty eight OECD = (Organization For Economic Co-operation and Development) countries would utilize the 5G connection to collect data management, and for further edge processing for real-time efficient, continuous data analysis and blockchain encryption technologies for Industrial Internet of Things (IoT) applications. According to scientists, the implementation of such solutions would maintain the integrity of transmitted data and have high process efficiency at the same time. Platforms integrating 5G connection with blockchain technologies for IoT are also widely discussed solutions in terms of advanced, intelligent, and efficient systems to optimize operations between machines, computers, and people [119,120]. In a broader perspective, the aforementioned solutions would allow the creation of “Smart Cities”, where modern technologies, including IoT, would be combined to solve communication, infrastructure, social, and pollution problems, aiming at improving the quality of life of metropolitan inhabitants [121,122,123].

Satellites are now successfully used to monitor weather events (storms, tornadoes, hurricanes, etc.), as well as to improve daily weather forecasts. Due to progress in satellite remote sensing, the investigation of climate changes and pollution monitoring with a high spatial resolution has become globally possible [124]. The main goal of cooperation between the 2018 NASA Health and Air Quality Applied Science Team (HAQAST) “Indicators” Tiger Team, and representatives of civil society was the development of satellite-based indicators for global assessment of air and climate pollution. The initial processing and documentation of the assumptions of the planned study, availability of data sets, and spatiotemporal variability, as well as development, analysis, and interpretation of the data sets requiring specialist technical knowledge before further implementation into environmental and public health surveillance purposes, were the biggest challenges to overcome [125].

Cooperation between the US Environmental Protection Agency (EPA), scientists from NASA, and the Smithsonian Astrophysical Observatory (SAO), and the National Oceanic and Atmospheric Administration (NOAA) aims to use the TEMPO (Tropospheric Emissions) satellite from 2022 to study air quality, taking into account the daily changes of O_3_, NO_2_, and other key elements in North America during the day in geostationary orbit, from an observation point approximately 22,000 miles above the Earth’s equator [126]. The GEMS Platform (Geostationary Environment Monitoring Spectrometer) [127] was designed to monitor air quality from the Earth’s geostationary orbit (GEO), with what the scientists say is “unprecedented spatial and temporal resolution”. The advancement of the GEMS system consists of UV-visible spectrometers with spectral resolution below nm (sub-nm spectral resolution) and advanced search algorithms, including the algorithm for measuring NO_2_ concentration, were developed, based on Differential Optical Absorption Spectroscopy (DOAS) [127].

With the development of technology, the quantity and quality of modern tools provide advanced possibilities for monitoring and analyzing the environment. Satellite observations and the development of artificial intelligence improve the comparability, and details of the collected data [128]. In the study conducted by van Donkelaar A et al., the combination of information from satellites, simulations and ground-based measurements enabled estimates of PM_2.5_ composition with promising accuracy (R2 = 0.57–R2 = 0.96) [129]. This analysis offered insight into the large spatiotemporal changes in PM_2.5_ composition over this period, driven by reductions in sulfate and organic matter. The presented approach could be readily adapted to other regions with PM_2.5_ ground monitoring networks, such as Europe or China. Annual PM_2.5_ composition estimates resulting from this effort are freely available as a public good from the Dalhousie University Atmospheric Composition Analysis Group Web site as version V4.NA.02 (North America) at http://fizz.phys.dal.ca/~atmos/martin/?page_id=140, accessed on 8 August 2022 or by contacting the authors [129].

Moreover, mention should be made of the soon to be added Multi-Angle Imager for Aerosols (MAIA) satellite with even more capabilities for compositional data estimation [130]. MAIA is a project created in response to NASA’s third Earth Venture Instrument (EVI) solicitation. Its primary objective is to explore the human health effects of exposure to different aerosol types. MAIA was created by a multidisciplinary science team that also includes epidemiologists, and is the first competitively selected NASA investigation aimed specifically and solely at societal benefit [130].

## 8. The Positive Effect of COVID-19-Induced Lockdown on Air Quality

Numerous scientific publications indicate the improvement of the air condition globally as a consequence of the lockdown forced by the SARS-CoV-2 pandemic [131,132]. A sharp decrease was observed in average concentrations of NO_2_, CO_2_, SO_2_, methane (CH_4_), and aerosols in India, China, and several large cities in Europe [133]. According to the data of the National Aeronautics and Space Administration (NASA), from February 2020 the global concentration of NO_2_ has decreased by nearly 20% compared to the year “without a pandemic” [134]. After analyzing the data, the highest decrease in concentration, by nearly 60%, was recorded by Madrid and Milan, and in New York by 45%. The presented data was compiled, based on mathematical models, and the results were presented in 2020 during the remote conference, *Conference for High—Performance Computing, Networking, Storage, and Analysis*. Similar conclusions regarding a significant decrease in air pollutant concentrations, in particular that of NO_2_ concentrations, were presented in March 2020 by the European Environment Agency (EEA) [135].

Calculations and conclusions on the impact of the pandemic on air quality in the five most affected European countries, in terms of concentrations of NO_2_, PM_2.5_, and PM_10_ recorded during subsequent phases of the lockdown in Great Britain, Spain, Sweden, Italy, and France, compared with values recorded for the years 2018–2019, were presented by a team of researchers from Lithuania [136]. The collected results clearly showed that the reduced industrial activity and lower mobility caused by the lockdown coincided with a reduction by about 20–40% in 2020 in the emission of selected pollutants: NO_2_, PM_2.5_, PM_10_. The results were also used to examine the economic dependencies in the context of changes in air pollution and the production index. Scientists point out that, while the global air quality has improved as a result of a lockdown that forced limitation of transport and minimized industrial activity, the economic effects after the SARS-CoV-2 pandemic require in-depth analysis, with particular emphasis on the impact of economic activity on air quality [136].

Air quality studies in the fifth largest world capitals, [137], indicated the highest decreases in PM_2.5_ concentrations in the capitals of America, Asia, and Africa with an overall 12% reduction in pollution during the COVID-19 lockdown. A significant improvement in air quality as a result of the lockdown, with decreases in concentrations of air pollutants NO_2_, PM_2.5_, PM_10_, O_3_, SO_2_, and CO, was also noted in India [138]. The presented comparative studies between 2019 and 2020 indicated that lowering the concentrations of the above-mentioned air pollutants significantly improved the values of the environmental impact (EI) and health risk (HR) indicators in cities such as Delhi, Bangalore, Hyderabad, and Kolkata.

Observations from research using geospatial technologies [139] confirmed that air pollution can be used to predict the severity of COVID-19 infections at the local or national level. One of the first nationwide studies, conducted in the US, confirmed that a slight increase in long-term exposure to PM_2.5_ resulted in a significant increase in COVID-19 mortality (an increase of 1 μg/m^3^ PM_2.5_ caused an 8% increase in COVID-19 mortality), and included twenty different confounding factors [140]. The results of the research conducted by Pansini et al. [141], in which the geographic nature of the infection was correlated with several annual and terrestrial air quality indicators (PM_10_, PM_2.5_,sulfur dioxide (SO_2_), carbon monoxide (CO), NO_2_ and ozone (O_3_), in the eight countries most severely affected by the pandemic, China, Iran, Italy, Spain, France, Germany, Great Britain, and the USA, indicated the possible geographic nature of SARS-CoV-2 infections [142]. Importantly, the relationship between air quality, the incidence, and induced mortality of COVID-19 appeared to be the highest in Italy. This relationship was also confirmed by other studies presented by scientists from this country [141,142,143]. The review presented in the *European Respiratory Journal* emphasizes that while short- and long-term exposure to air pollution may be an important factor toSARS-CoV-2 transmission and mortality due to COVID-19, the links between air pollution, SARS-CoV-2 infection, and severe acute respiratory syndrome require further analysis [144]. A new index, defined by the *index c* (contagions), has been proposed to quantify the environmental risk exposure of economic areas to new epidemics, even before their occurrence (*ex-ante*) [145]. In the presented calculation model, air pollution was the first factor (before atmospheric stability expressed by wind speed, population density, and human respiratory system disorders determined by the mortality rate due to cancer of the trachea, bronchi, and lungs), determining the spread of infectious diseases, using the example of COVID-19. According to the researchers, the model, using Index c, will help to prevent possible epidemics and pandemics in the future.

## 9. Conclusions

Despite numerous publications focusing on the relationship between the development of T1DM and air quality, knowledge of the health consequences of exposure to air pollution, including the presence of heavy metals, is still an important research problem requiring further examination. Nevertheless, air pollution is undoubtedly a factor attributing to premature deaths. The main focus of the current review was the risk of T1DM development associated with exposure to polluted air. Modern and more adequate methods for air pollution monitoring were also introduced, with a special emphasis on micro-sensors, mobile and unmanned measuring platforms, satellites, and innovative approaches of IoT, 5G connections, and Blockchain technologies.

While the data about air pollution continues to grow worldwide with new technologies and more sensitive sensors, the association of the particular composition of air pollution with autoimmune diseases, such as T1DM, should also be explored in detail. Although the specific molecular mechanism(s) behind the link between air pollution exposure and a higher risk of diabetes and metabolic dysfunction is yet unknown, the available data indicates air pollution-induced inflammation and oxidative stress as a significant pathway. Improved understanding of environmental factors favorable to the development of different civilizational diseases, and autoimmunity, including T1DM, would be beneficial for further disease prevention, and implementation of emission control legislation.

## Figures and Tables

**Figure 1 antioxidants-11-01908-f001:**
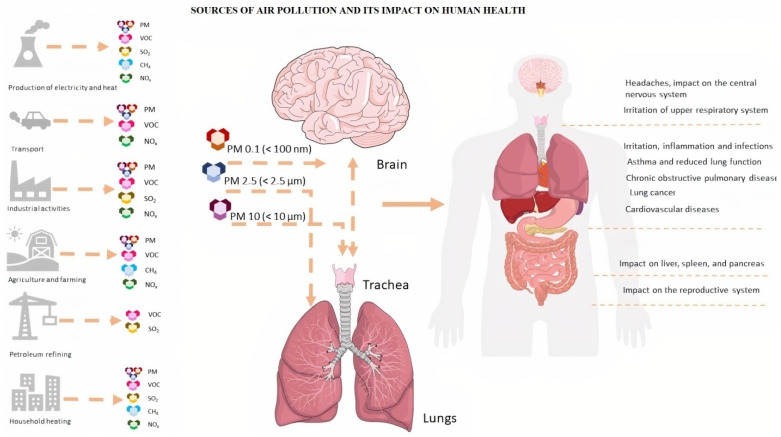
Major sources of air pollution and its negative impact on human health. The figure prepared by Marta Jaskulak on the basis of literature [4,5,9,24].

**Figure 2 antioxidants-11-01908-f002:**
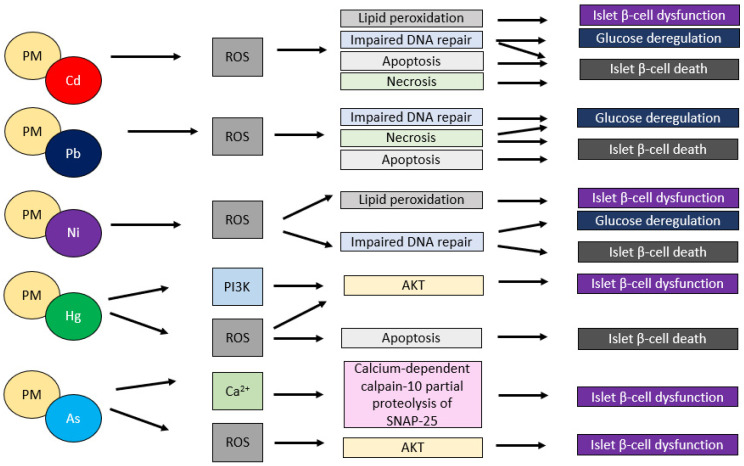
Schematic representation of intracellular signaling leading to PM-adsorbed toxic metals induced β-cell dysfunction. The figure prepared by Marta Jaskulak on the basis of literature [2,3,22].

**Table 1 antioxidants-11-01908-t001:** Air pollution and type 1 diabetes.

No	Reference	Air Pollutants	Main Findings & Health Effects
1.	[6]	PM_10_, PM_2.5_	Air pollution was associated with higher HbA1c levels and increased risk of severe hypoglycemia in people with T1DM, consequently indicating a higher risk of diabetes complications.
2.	[23]	PM_10_	An analysis of linear regression revealed a relation between the number of new T1DM cases and the mean annual concentration of PM_10_ for the year 2016 (*p* < 0.001). However, there was no relationship observed between the number of new cases of T1DM and the mean annual concentration of PM_10_ in the air in the Lubelskie Voivodeship in 2015.
3.	[26]	PM_10_, NO_2_, NO_x_, SO_2_, CO	High exposure to gaseous pollutants and particulate matter in ambient air may be one of the factors contributing to the risk of developing T1DM in children.
4.	[29]	PM_10_, NO_2_, SO_2_, CO, O_3_	The associations between different markers of air pollutants and markers of inflammation, oxidative stress, and insulin resistance.
5.	[30]	NO_2_, PM_10_, PM_2.5._	Traffic-related air pollution may increase the risk of insulin resistance.
6.	[31]	PM_2.5_	Youth with type 1 diabetes may be at increased risk of air pollution-related inflammation.
7.	[34]	PM_10_, NO_2_, O_3_-AOT40	No adverse effect of PM_10_, NO_2_, and O_3_-AOT on HbA1c levels in type 1 diabetes.
8.	[35]	PM_10_, NO_2_, O_3_-AOT	No significant associations between O_3_-AOT, PM_10_ and NO_2_ and insulin dose. An inverse association between O_3_-AOT and HbA1c.
9.	[36]	PM_10_, NO_2_, O_3_-AOT	Notably higher HbA1c levels with higher PM_10_ and NO_2_ concentration, an inverse association between O_3_-AOT and HbA1c, differences in HbA1c between air pollution quartiles were small but statistically significant.
10.	[37]	NO_x_, O_3_	Positive association between air pollution exposure and T1D. Living in an area with elevated levels of air pollution during pregnancy may be a risk factor for T1D in offspring.
11.	[39]	NO_2_, PM_2.5_, O_3_, O_X_ = (O_3_ + NO_2_)	O_3_ exposures during a critical period of development were associated with an increased risk of pediatric diabetes incidence.
12.	[40]	PM_10_	Chronic PM exposure by the oral route during perinatal life in rats led to glucose dyshomeostasis in male offspring, both in early and later life, and ambiance with poor air quality, mainly where traffic is dense, can contribute to an increase in incidence of metabolic disease.
13.	[42]	SO_2_, SO_4_, O_3_	Cumulative exposure to air pollution and sulfate in air may predispose the development of type 1 diabetes in children.
14.	[43]	PM_10_, NO_x_, CO, O_3_	PM_10_ exposure significantly affects the incidence of T1D.
15.	[44]	PM_10_, PM_2.5_, NO_2_	High exposure to the traffic-related air pollutants PM_10_, NO_2_, and possibly PM_2.5_, accelerate the manifestation of T1D, but only in very young children.
16.	[45]	PM_2.5_	Exposure to PM_2.5_ during pregnancy is associated with increased levels of cord plasma insulin at birth.
17.	[46]	PM, heavy metals adsorbed on PM	The unique geochemical profile of Sardinia, with particular density of heavy metals, lead to the assumption that exposure of the population to heavy metals could also affect T1D incidence, due to their presence in PM.
18.	[47]	NO_2_, SO_2_, O_3_, PM_10_ and PM_2.5_	Exposure to ozone and solar radiation during gestation were both associated with T1DM in offspring, although at borderline significance.
19.	[48]	PM_2.5_	Exposure to PM_2.5_ in the first and second trimesters was related to gestational diabetes mellitus.

**Table 2 antioxidants-11-01908-t002:** The exposure to selected heavy metals and the possible risk of developing type 1 diabetes.

	Reference	Exposure to Selected Heavy Metals	Health Effects
1	[25]	Chromium (Cr), Copper (Cu), Iron (Fe), Manganese (Mn), Mercury (Hg), Nickel (Ni), Lead (Pb), Selenium (Se), Zinc (Zn)	Type 1 diabetes was found to be associated with Cr (*p* = 0.02), Mn (*p* < 0.001), Ni (*p* < 0.001), Pb (*p* = 0.02), Zn (*p* < 0.001). In type 1 diabetes, there was a positive correlation between Pb and age (*p* < 0.001, ρ = 0.400), and Pb and BMI (*p* < 0.001, ρ = 0.309), and a negative correlation between Fe and age (*p* = 0.002, ρ = −0.218).
2	[29]	Selenium (Se), Zinc (Zn), Magnesium (Mg), Copper (Cu),	Children with T1D, especially poorly controlled cases, had low serum Se, Zn, Mg, Cu, GSH, and GPx. Low serum Se in diabetic children may affect the erythrocyte GSH-GPx system.
3	[30]	Copper, (Cu), Copper (-Cu), Zinc (Zn), Manganese (Mn), Selenium (Se), Iron (Fe)	Total copper and ceruloplasmin levels were higher in persons with T1D compared to healthy controls. Manganese, Zinc, and Selenium were significantly lower in TD1 patients. Iron did not differ between the TD1 patients and healthy controls.
4	[60]	Arsenic (As), Aluminum (Al), Cadmium (Cd), Lithium (Li), Mercury (Hg), Lead (Pb)	Exposure to toxic metals during pregnancy might be one among several contributing environmental factors to the disease process.
5	[63]	Copper (Cu), Zinc (Zn), Selenium (Se), Chromium (Cr), Iron (Fe)	Zn was not correlated with HbA1c and FPG. In males, Cr was positively correlated with HDL *p* = 0.0079) and with fasting plasma glucose (*p* = 0.0149), particularly in males (*p* = 0.0038). Copper was significantly correlated with HbA1c% in males (*p* = 0.0155).
7	[66]	Selenium (Se), Zinc (Zn), Magnesium (Mg), Copper (Cu)	Plasma magnesium concentration was decreased in T1DM. HbA1c correlated with plasma concentrations of magnesium (negatively, in both sexes together in T1DM and T1DM males), copper (positively, in T1DM males), selenium (positively, in both sexes together in T1DM), and with zinc/copper ratio (negatively, in both sexes together in T1D).
8	[67]	Zinc (Zn)	Plasma zinc levels were significantly higher in type-1 diabetes mellitus patients, while plasma copper and magnesium levels were not significantly altered. No effect of sex, glycemic control, or presence of microalbuminuria could be demonstrated on plasma concentration of trace elements in type-1 diabetes mellitus patients.
9	[69]	Zinc (Zn)	Kidney tissue antioxidant enzyme activities, which were significantly impaired in the untreated diabetic group, were reversed in zinc treated diabetic groups, thus showing the beneficial effect of Zn treatment in diabetes via its antioxidative effects.
10	[70]	Cadmium (Cd)	A significantly lower glucose elevation rate was observed in the insulin response test after an insulin-induced decrease in glucose level in Cd-exposed animals. Decreased corticosterone levels, together with increased E2 and reduced P4 levels, were some of the hallmark changes in the serum hormone profile of Cd-exposed animals.
11	[71]	Magnesium (Mg), Selenium (Se), Zinc (Zn), Manganese (Mn), and Copper (Cu)	Compared with the control group, T1DM children had lower plasma levels of Mg and Zn and higher levels of Cu.
12	[72]	Chromium (Cr), Cobalt (Co), Nickel (Ni)	Patients who suffered from 1 type diabetes mellitus and had a high risk for life level of glycemic control had considerable polideficiency of cobalt, nickel, and chromium in serum. An increasing level of chromium was recorded only in the erythrocytes. It was found that the duration of type 1 diabetes mellitus influenced the levels of cobalt and nickel in serum mostly, while the level of glycemic control influenced the chromium content.
13	[73]	Chromium (Cr) and Iron (Fe)	The urinary Cr level in T1D was the highest of all, which significantly exceeded those of the T2D groups, with and without complications. No significant differences in serum Fe levels were found among all groups. The urinary Fe level of T1D was significantly increased (<0.05). These results suggested the potential role of Cr and Fe in diabetes should receive attention.
14	[74]	Chromium picolinate (CrPic)	Eight weeks of CrPic supplementation was found to repair renal function and reverse renal pathological changes (renal interstitial fibrosis and glomerular sclerosis) in diabetic nephropathy rats by an antioxidative stress mechanism, as well as by inhibiting TGF-β1 and Smad2/3 expressions. CrPic could, thus, be used as a nutrient to prevent diabetic nephropathy.

**Table 3 antioxidants-11-01908-t003:** Air pollution and chronic vascular complications.

	Authors	Air Pollution	Chronic Vascular Complications
1.	[6]	PM_10_, PM_2.5_	A higher risk of diabetes complications.
2.	[82]	PM with Cd and Pb	The recognition of Pb and Cd as contaminants that are common in PM increase atherosclerotic cardiovascular risk in a dose-dependent manner.
3.	[83]	PM_2.5_ with adsorbed heavy metals	Nine metals in PM_2.5_ were significantly associated with cardiovascular disease mortality: Pb, Cd, As, Se, Sb, Th, Al, Fe. Specific control measures, aimed at the emission sources, should be taken to reduce the cardiovascular disease mortality risk of PM_2.5_.
4.	[85]	Pb-PM_2.5_	Pb-bound PM_2.5_ was the main contributor to season-dependent health risks for children. Pb-containing PM_2.5_ caused ROS-mediated myocardial toxicity.
5.	[87]	PM_10_, PM_2.5_	In a European cohort study of 100,166 patients, followed, on average, for 11.5 years, a 100 ng/m^3^ increase in PM_10_ and a 50 ng/m^3^ increase in PM_2.5_ resulted in 6% and 18% increase in coronary events.
6.	[95]	PM	higher risk of diabetic retinopathy
7.	[97]	PM_2.5_, PM_10_, NO_2_, SO_2_ and O_3_	An increased risk of obstruction of the central retinal artery, especially in patients with diabetes or hypertension, and people over 65 years of age.
8.	[98]	PM_2.5_	The burden of diabetes and of kidney disease attributable to PM_2.5_ pollution.
9.	[106]	NO_2_	Even low levels of traffic-related air pollutants linked to atherosclerotic plaque burden.
10.	[107]	PM_2.5_ and NO_2_	Long-term PM_2.5_ and NO_2_ exposures were associated with higher blood pressure.
11.	[108]	PM_2.5_	PM_2.5_ was associated with a higher prevalence of diabetes, hypertension, and being overweight in a less-educated population.

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
