# Peer review of "Air Pollution, Oxidative Stress, and the Risk of Development of Type 1 Diabetes"

_antioxidants, 2022, doi:10.3390/antiox11101908_

Round 1
Reviewer 1 Report
Dear authors,
After the review report, I have several comments: how was realized figures 1, 2; copyright?!; in tables 1, 2 and 3, is not necessary the name of the authors; in this paper, the authors do not make any relation with microbiota, that is an important detail in this pathologies. Interaction with oxidative stress is essential for pathology clinical manifestation thus you should add new findings/data about human microbiota response to Exposure to different metals and oxidative stress.
Best regards!
Author Response
Response to reviewers’ comments:
Thank you very much for reviewing our manuscript. We wish to express our appreciation to the reviewers for their useful comments, which have helped us to improve our manuscript. According to the suggestions, we have revised our manuscript and its final version is enclosed. Please find attached a point-by-point response to the reviewer’s concerns. We hope that you find our responses satisfactory and that the manuscript is now acceptable for publication.
Response to reviewer 1:
Reviewer wrote:
Dear authors,
After the review report, I have several comments: how was realized figures 1, 2; copyright?!; in tables 1, 2 and 3, is not necessary the name of the authors; in this paper, the authors do not make any relation with microbiota, that is an important detail in this pathologies. Interaction with oxidative stress is essential for pathology clinical manifestation thus you should add new findings/data about human microbiota response to Exposure to different metals and oxidative stress.
Response:
Thank you for your kind review. We revised the manuscript with regards to all of your comments. Figures 1 and 2 are original figures created by a co-author (dr Marta Jaskulak) as a collective visualization of more than one literature point – we added the references on which the figures are now included in the figure captions.
We agree that the names of authors are unnecessary in the tables and it will make the tables “wider” and more readable – the entire column with authors names has been removed.
Microbiota comment: We fully acknowledge the role of gut microbiota in predisposition and onset of type 1 diabetes. However, the microbiome would deserve its own manuscript due to its complexity. However, as suggested by the reviewer, in the present review, it has been added to item 5 as: Effects of heavy metals, particulate matter on the oxidative stress, and gut microbiome.
In addition to oxidative stress, the influence of heavy metals on the gut microbiome also plays a role. Previous studies have shown that exposure to heavy metals adsorbed on PM has a significant effect on the intestinal microbiome, leading to dysbiosis of the physiological bacterial flora [89], [90]. To determine how particulate matter affect the intestines and gut microbiome, wild-type (WT) mice were exposed orally to PM10 for 7 or 14 days. In turn, to assess longer term effects of exposure, IL10 deficient (−/−) mice were subjected to the same treatment for 35 days [89]. WT mice exposed to PM10 for a short duration had alteration in immune gene expression, increased gut leakiness, enhanced pro-inflammatory cytokine secretion into the small intestine. In other words, a significant increase in IL-17, IL-1β, TNFα, IL-12 and IL-13 was seen in the colon of IL-10−/− mice treated with PM10. PM10 induced increased IL-17 and IL-13 in the colons of WT mice, and these animals demonstrated significant changes in the relative amounts of Bacteroidetes spp., Firmicutes spp., and Verrucomicrobia spp. [89].
Arsenic is ubiquitous in nature, highly toxic, and in some countries contamination in drinking water has been shown to be above the WHO and EPA recommended limit of 10 μg / L [90]. In a recent study, the authors examined stool samples (n = 42) collected from members of the Mahuawa (n = 20) and Ghanashyampur (n = 22) communities in southern Nepal [90]. The 16S rRNA gene was amplified from fecal samples using Illumina-tag PCR and subject to high-throughput sequencing to generate the bacterial community structure of each sample. Bioinformatics analysis and multivariate statistics were conducted to identify if specific fecal bacterial assemblages and predicted functions were correlated with urine arsenic concentration. The conducted results revealed unique assemblages of arsenic volatilizing and pathogenic bacteria positively correlated with increased arsenic concentration in individuals within the two respective communities. Additionally, they observed that commensal gut bacteria negatively correlated with increased arsenic concentration in the two respective communities. The study authours has revealed that arsenic has the potential to debilitate healthy humans by contributing to disorders like heart and liver cancers and diabetes, and it has already been shown to contribute to serious diseases and disorders, including skin lesions, gangrene and several types of skin, renal, lung, and liver cancers in disadvantaged areas of the world like Nepal [90].
In other studies, the effect of lead on bacterial microflora in experimental mice was presented. Non-agouti (a/a) offspring derived from Avy/a male mice bred to a/a female mice exposed from gestation through lactation to Pb (32 ppm in the drinking water) undergo shifts in gut microbiota populations with Bacteroidetes and Firmicutes inversely associated with maternal Pb exposure. Cultivable aerobes decreased but anaerobes increased in the Pb-exposed offspring. Intestinal flora changes were associated with an increase in adult body weight in males but not females[51].
Interesting research was presented by the team of Griggs JL et al., [91]. In recent studies, the authors explored the relation between arsenic exposure and changes in the composition of the gut microbiome and in arsenic bioaccessibility. Here, they used simulated GI model system (GIMS) containing a stomach, small intestine, colon phases and microorganisms obtained from mouse feces (GIMS-F) and cecal contents (GIMS-C) to assess whether exposure to arsenic-contaminated soils affect the gut microbiome and whether composition of the gut microbiome affects arsenic bioaccessibility. Soils contaminated with arsenic did not alter gut microbiome composition in GIMS-F colon phase. In contrast, arsenic exposure resulted in the decline of bacteria in GIMS-C, including members of Clostridiaceae, Rikenellaceae, and Parabacteroides. Arsenic bioaccessibility was greatest in the acidic stomach phase of GIMS (pH 1.5-1.7); except for GIMS-C colon phase [91]. In other study, in 6 to 8-week-old C57Bl/6 Tac male mice exposed for 2, 5, or 10 weeks to 0, 10, or 250 ppb arsenic, time and dose-dependent effects on the gut microbial community were found, especially for Bacteroidetes and Firmicutes [92]. Arsenic-treatment removed the bacterial biofilm residing along the mucosal lining and altered the diversity and abundance of microorganisms with bacterial spores increasing and intracellular inclusions reduced with the 250 ppb dose [92].
In the present manuscript, only a few examples are given of the effects of heavy metals from soil, water, and air on the gastrointestinal microbiome, as this issue is beyond the subject covered in the manuscript.
[89] Kish, L., Hotte, N., Kaplan, G. G., Vincent, R., Tso, R., Ganzle, M. Environmental particulate matter induces murine intestinal inflammatory responses and alters the gut microbiome. PLoS ONE, 20138:e62220. doi: 10.1371/journal.pone.0062220
[90] Brabec JL, Wright J, Ly T, Wong HT, McClimans CJ, Tokarev V, Lamendella R, Sherchand S, Shrestha D, Uprety S, Dangol B, Tandukar S, Sherchand JB, Sherchan SP. Arsenic disturbs the gut microbiome of individuals in a disadvantaged community in Nepal. Heliyon. 2020 Jan 31;6(1):e03313. doi: 10.1016/j.heliyon.2020.e03313. eCollection 2020 Jan.
[51] Wu J, Wen XW, Faulk C, Boehnke K, Zhang H, Dolinoy DC, Xi C. Perinatal Lead ExposureAlters Gut Microbiota Composition and Results in Sex-specific Body weight Increases in Adult Mice.Toxicol Sci.2016, Jun;151(2):324-33. doi: 10.1093/toxsci/kfw046. Epub 2016 Mar 8.
[91] Griggs JL, Chi L, Hanley NM, Kohan M, Herbin-Davis K, Thomas DJ, Lu K, Fry RC, Bradham KD. Bioaccessibility of arsenic from contaminated soils and alteration of the gut microbiome in an in vitro gastrointestinal model.EnvironPollut. 2022 Sep 15;309:119753. doi: 10.1016/j.envpol.2022.119753. Epub 2022 Jul 11.
[92] Dheer R., Patterson J., Dudash M., Stachler E. N., Bibby K. J., Stolz D. B., Shiva S, Wang Z, Hazen SL, Barchowsky A, Stolz JF. Arsenic induces structural and compositional colonic microbiome change and promotes host nitrogen and amino acid metabolism. Toxicol Appl Pharmacol. 2015 Dec 15;289(3):397-408. doi: 10.1016/j.taap.2015.10.020. Epub 2015 Oct
Again, we thank you for your valuable remarks. We have done our best to ensure that the corrected version of the manuscript can be read clearly, and is concise and readable for whom it may concern.
Reviewer 2 Report
This manuscript is fine as far as it goes, but where is the point? The review needs to add a final section with Conclusions and Recommendations that lists the major knowledge gaps and the research needs identified during the review. In addition, here are specific comments/edits needed I noted:
Abstract
Pg 1, line 20“available data indicates to” should be “available data indicate”
Introduction
Pg 1, line 40: “air pollution caused 4.9 million deaths” should read “outdoor air pollution caused 4.9 million deaths”
Methods
Pg. 3, line 105: “Specific heavy metals that have been reported to be associated with particulate matter were also added to searches including (but, not limited to)” should have carefully considered coal combustion tracers, Arsenic and Selenium and oil tracer Vanadium (a transition metal). Did it?
Pg 3. Line 108 the primary focus on “recent publication year” seems like it will inappropriately leave out many important contributions to the literature prior to that time period.
Pg. 13, line 455: Ref 46 does not support the statement made about metals.
Perhaps you meant something like:
Maciejczyk P et al. The Role of Fossil Fuel Combustion Metals in PM2. 5 Air Pollution Health Associations. Atmosphere. 2021. 12 (9), 1086.
Pg. 17. The discussion of satellites should mention the satellite-based particulate matter databases available for use in air pollution health studies around the globe, and cite how this can address the dearth of air pollution monitoring data being collected in the developing world and in rural places around the globe. E.G. see:
Van Donkelaar, A., Martin, R.V., Li, C., Burnett, R.T., 2019. Regional Estimates of Chemical Composition of Fine Particulate Matter Using a Combined Geoscience- Statistical Method with Information from Satellites, Models, and Monitors. Environ. Sci. Technol. 53, 2595–2611. https://doi.org/10.1021/acs.est.8b06392.
In addition, mention the soon to be added MAIA satellite with even more capabilities for compositional data estimation:
https://www.nae.edu/260862/MAIA-Opportunities-and-Challenges-Associated-with-a-Small-CostCapped-Satellite-Mission
Author Response
Response to reviewer 2:
Reviewer wrote:
This manuscript is fine as far as it goes, but where is the point? The review needs to add a final section with Conclusions and Recommendations that lists the major knowledge gaps and the research needs identified during the review.
Response:
Thank you for your review and valuable comments. We revised our manuscript in accordance to all your remarks. You can see the changes below and in the tracked version of the manuscript. As suggested, the conclusions were expanded. You can see the added text down below:
While the data about the air pollution continues to grow worldwide with new technologies and, more sensitive sensors, the association of particular composition of air pollution with autoimmune diseases such as T1DM should also be explored in detail. Although the specific molecular mechanism(s) behind the link between air pollution exposure and a higher risk of diabetes and metabolic dysfunction is yet un-known, the available data indicates to air pollution-induced inflammation and oxidative stress as a significant pathway. Improved understanding of environmental factors favorable to the development of different civilizational diseases, and autoimmune including T1DM would be beneficial for further disease prevention, and implementation of emission control legislation.
Reviewer wrote:
In addition, here are specific comments/edits needed I noted:
Abstract
Pg 1, line 20“available data indicates to” should be “available data indicate”
Introduction
Pg 1, line 40: “air pollution caused 4.9 million deaths” should read “outdoor air pollution caused 4.9 million deaths”
Response:
Thank you for those remarks, the Pg 1 line 20 was fixed to “indicate”, and Pg1, line 40 the word “outdoor” was added.
Reviewer wrote:
Methods
Pg. 3, line 105: “Specific heavy metals that have been reported to be associated with particulate matter were also added to searches including (but, not limited to)” should have carefully considered coal combustion tracers, Arsenic and Selenium and oil tracerVanadium (a transition metal). Did it?
Response:
Thank you for this comment, yes, indeed, the link with arsenic, selenium and vanadium was also used during the collection of bibliography. As suggested by the Reviewer, more information has been added to the revised version of the manuscript i.e.
Page 4-5
Since heavy metals like cadmium (Cd), (Zn), lead (Pb), copper (Cu), and nickel (Ni), can adsorb to the PM surface and contribute to the toxic consequences of PM exposure, the Scientists' attention is also paid to the importance of trace elements, heavy metals, and metalloids (e.g. Se or Ar) in the development of T1DM [29],[30]. Alghobashy, A. et al., [29] in study showed that the serum Se, Zn, Mg, and Cu were significantly lower in the diabetic group in comparison to the control group. This agrees with Özenç et al., [8] who found lower serum Se and Zn and normal serum Cu levels in children with T1D in comparison to controls [8]. They explained the low serum Se level in patients with T1D might be due to its consumption by the increased activity of the antioxidant GSH-GPx system in order to reduce the free radicals produced by increased oxidative stress [29].
Chiu YM et al., examined associations among prenatal PM2.5, maternal antioxidant intake, and childhood wheeze in an urban pregnancy cohort (n = 530). Daily PM2.5 exposure over gestation was estimated using a satellite-based spatiotemporally resolved model. Mothers completed the modifie food frequency questionnaire. Average energy-adjusted percentile intake of β-carotene, vitamins (A, C, E), and trace minerals (zinc, magnesium, selenium) constituted an antioxidant index (AI) [24]. Higher AI was associated with decreased wheeze in Blacks (OR = 0.37 (0.19–0.73), per IQR increase). BDLIMs identified a sensitive window for PM2.5 effects on wheeze among boys born to Black mothers with low AI (at 33–40 weeks gestation; OR = 1.74 (1.19–2.54), per µg/m3 increase in PM2.5). Relationships among prenatal PM2.5, antioxidant intake, and child wheeze were modified by race/ethnicity and sex [24].
Page 7-8
Vandium (V) is the 22nd most abundant element on earth (0.013% w/w), and it is widely distributed in all organisms. In humans, the vanadium content in blood plasma is around 200 nM, while in tissues is around 0.3 mg/kg and mainly found in bones, liver, and kidney. In vertebrates, vanadium enters the organism principally via the digestive and respiratory tracts through food ingestion and water, air inhalation [58]. So far, the authors have shown that that vanadium administered in the drinking water to streptozotocin (STZ) diabetic rats restored elevated blood glucose to normal. Subsequent studies have shown that vanadyl sulfate can lower elevated blood glucose, cholesterol and triglycerides in a variety of diabetic models. In the BB diabetic rat, a model of insulin-dependent diabetes, vanadyl sulfate lowered the insulin requirement by up to 75% [58].
Galvez-Fernandez et al. [59] analyzed the association of 11 metals ( urine antimony, arsenic, barium, cadmium, chromium, cobalt, molybdenum, vanadium, and plasma copper, selenium and zinc) with metabolic patterns, and the interacting role of candidate genetic variants, in 1145 participants, a population-based sample from Spain. Exposures to cobalt, plasma copper, selenium, zinc, arsenic but not to vanadium were associated with several metabolic patterns involved in chronic disease [59]. Vanadium compounds have been primarily investigated as potential therapeutic agents for the treatment of various major health issues, including cancer, atherosclerosis, and diabetes. The translation of vanadium-based compounds into clinical trials and ultimately into disease treatments remains hampered by the absence of a basic pharmacological and metabolic comprehension of such compounds [58],[60 ].
Line 518, page 14-15
Arsenic is ubiquitous in nature, highly toxic, and in some countries contamination in drinking water has been shown to be above the WHO and EPA recommended limit of 10 μg / L [90]. In a recent study, the authors examined stool samples (n = 42) collected from members of the Mahuawa (n = 20) and Ghanashyampur (n = 22) communities in southern Nepal [90]. The 16S rRNA gene was amplified from fecal samples using Illumina-tag PCR and subject to high-throughput sequencing to generate the bacterial community structure of each sample. Bioinformatics analysis and multivariate statistics were conducted to identify if specific fecal bacterial assemblages and predicted functions were correlated with urine arsenic concentration. The conducted results revealed unique assemblages of arsenic volatilizing and pathogenic bacteria positively correlated with increased arsenic concentration in individuals within the two respective communities. Additionally, they observed that commensal gut bacteria negatively correlated with increased arsenic concentration in the two respective communities. The study authours has revealed that arsenic has the potential to debilitate healthy humans by contributing to disorders like heart and liver cancers and diabetes, and it has already been shown to contribute to serious diseases and disorders, including skin lesions, gangrene and several types of skin, renal, lung, and liver cancers in disadvantaged areas of the world like Nepal [90].
Interesting research was presented by the team of Griggs JL et al., [91]. In recent studies, the authors explored the relation between arsenic exposure and changes in the composition of the gut microbiome and in arsenic bioaccessibility. Here, they used simulated GI model system (GIMS) containing a stomach, small intestine, colon phases and microorganisms obtained from mouse feces (GIMS-F) and cecal contents (GIMS-C) to assess whether exposure to arsenic-contaminated soils affect the gut microbiome and whether composition of the gut microbiome affects arsenic bioaccessibility. Soils contaminated with arsenic did not alter gut microbiome composition in GIMS-F colon phase. In contrast, arsenic exposure resulted in the decline of bacteria in GIMS-C, including members of Clostridiaceae, Rikenellaceae, and Parabacteroides. Arsenic bioaccessibility was greatest in the acidic stomach phase of GIMS (pH 1.5-1.7); except for GIMS-C colon phase [91]. In other study, in 6 to 8-week-old C57Bl/6 Tac male mice exposed for 2, 5, or 10 weeks to 0, 10, or 250 ppb arsenic, time and dose-dependent effects on the gut microbial community were found, especially for Bacteroidetes and Firmicutes [92]. Arsenic-treatment removed the bacterial biofilm residing along the mucosal lining and altered the diversity and abundance of microorganisms with bacterial spores increasing and intracellular inclusions reduced with the 250 ppb dose [92].
Reviewer wrote:
Pg 3. Line 108 the primary focus on “recent publication year” seems like it will inappropriately leave out many important contributions to the literature prior to that time period.
Response:
We apologize for our inattention as it should read: "recent publication years"
Reviewer wrote:
Pg. 13, line 455: Ref 46 does not support the statement made about metals.
Perhaps you meant something like:
Maciejczyk P et al. The Role of Fossil Fuel Combustion Metals in PM2.5 Air Pollution Health Associations. Atmosphere. 2021. 12 (9), 1086.
Response: Thank you for this comment. The Maciejczyk et al., reference has been added in the suggested place as reference no 85.
Reviewer wrote:
Pg. 17. The discussion of satellites should mention the satellite-based particulate matter databases available for use in air pollution health studies around the globe, and cite how this can address the dearth of air pollution monitoring data being collected in the developing world and in rural places around the globe. E.G. see:
Van Donkelaar, A., Martin, R.V., Li, C., Burnett, R.T., 2019. Regional Estimates of Chemical Composition of Fine Particulate Matter Using a Combined Geoscience- Statistical Method with Information from Satellites, Models, and Monitors. Environ. Sci. Technol. 53, 2595–2611. https://doi.org/10.1021/acs.est.8b06392.
In addition, mention the soon to be added MAIA satellite with even more capabilities for compositional data estimation:
https://www.nae.edu/260862/MAIA-Opportunities-and-Challenges-Associated-with-a-Small-CostCapped-Satellite-Mission
Response:
Thank you for your remark.In the revised version of the manuscript, we added a information e.i.
In the study van Donkelaar A et al., showed the combination of information from satellites, simulations and ground-based measurements enabled estimates PM2.5 composition with promising accuracy (R2 = 0.57 – R2 = 0.96) [129]. This analysis offered insight into the large spatiotemporal changes in PM2.5 composition over this period, driven by reductions in sulfate and organic matter. The approach presented could be readily adapted to other regions with PM2.5 ground monitoring networks, such as Europe or China. Annual PM2.5 composition estimates resulting from this effort are freely available as a public good from the Dalhousie University Atmospheric Composition Analysis Group Web site as version V4.NA.02 (North America) at http://fizz.phys.dal.ca/~atmos/martin/?page_id=140, or by contacting the authors [129].
Moreover, mention should be made of the soon to be added Multi-Angle Imager for Aerosols (MAIA) satellite with even more capabilities for compositional data estimation [130]. MAIA is a project created in response to NASA’s third Earth Venture Instrument (EVI) solicitation. Its primary objective is to explore the human health effects of exposure to different aerosol types. MAIA is created by a multidisciplinary science team that also includes epidemiologists, and is the first competitively selected NASA investigation aimed specifically and solely at societal benefit [130].
Again, we thank you for your valuable remarks. We have done our best to ensure that the corrected version of the manuscript can be read clearly, and is concise and readable for whom it may concern.
Round 2
Reviewer 1 Report
No other comments.
Reviewer 2 Report
Revision is responsive to my concerns.